# Enhancing combinatorial optimization with classical and quantum generative models

Javier Alcazar[1,2], Mohammad Ghazi Vakili [1,3,4], Can B. Kalayci [1,5] &
Alejandro Perdomo-Ortiz [1] ✉

Devising an efficient exploration of the search space is one of the key challenges in the design of combinatorial optimization algorithms. Here, we introduce the Generator-Enhanced Optimization (GEO) strategy: a framework that leverages any generative model (classical, quantum, or quantum-inspired) to solve optimization problems. We focus on a quantum-inspired version of GEO relying on tensor-network Born machines, and referred to hereafter as TN-GEO. To illustrate our results, we run these benchmarks in the context of the canonical cardinality-constrained portfolio optimization problem by constructing instances from the S&P 500 and several other financial stock indexes, and demonstrate how the generalization capabilities of these quantum-inspired generative models can provide real value in the context of an industrial application. We also comprehensively compare state-of-the-art algorithms and show that TN-GEO is among the best; a remarkable outcome given the solvers used in the comparison have been fine-tuned for decades in this real-world industrial application. Also, a promising step toward a practical advantage with quantum-inspired models and, subsequently, with quantum generative models

Along with machine learning and the simulation of materials, combinatorial optimization is one of top candidates for practical quantum advantage. That is, the moment where a quantum-assisted algorithm outperforms the best classical algorithms in the context of a real-world application with a commercial or scientific value. There is an ongoing portfolio of techniques to tackle optimization problems with quantum subroutines, ranging from algorithms tailored for quantum annealers (e.g., refs. 1,2), gate-based quantum computers (e.g., refs. 3,4) and quantum-inspired (QI) models based on tensor networks (e.g., ref. 5).

Regardless of the quantum optimization approach proposed to date, there is a need to translate the real-world problem into a polynomial unconstrained binary optimization (PUBO) expression – a task which is not necessarily straightforward and that usually results in an overhead in terms of the number of variables. Specific real-world use cases illustrating these PUBO mappings are depicted in refs. 6 and [7].

Therefore, to achieve practical quantum advantage in the near-term, it would be ideal to find a quantum optimization strategy that can work on arbitrary objective functions, bypassing the translation and overhead limitations raised here.

In our work, we offer a solution to these challenges by proposing a generator-enhanced optimization (GEO) framework which leverages the power of (quantum or classical) generative models. This family of solvers can scale to large problems where combinatorial problems become intractable in real-world settings. We present the main results where we highlight the different features of GEO by performing a comparison with alternative solvers, such as Bayesian optimizers, and generic solvers, like simulated annealing. In the case of the specific real-world, large-scale application of portfolio optimization, we compare against the state-of-the-art (SOTA) optimizers and show the competitiveness of our approach.

[1]Zapata Computing Canada Inc., 25 Adelaide St E, Suite 1500, Toronto, ON M5C 3A1, Canada. [2]Acadian Asset Management LLC, 24 King William St, London EC4R 9AT, England. [3]Department of Chemistry, University of Toronto, Toronto, ON M5G 1Z8, Canada. [4]Department of Computer Science, University of Toronto, Toronto, ON M5S 2E4, Canada. [5]Department of Industrial Engineering, Pamukkale University, Kinikli Campus, 20160 Denizli, Turkey. ✉ e-mail: aperdomo@post.harvard.edu

## Results

### Preliminaries

Here are more salient highlights that make the proposed approach advantageous over other available solvers:

- *It leverages the power of generative models:* The essence of the solver is that it is aiming to unveil non-obvious structure in the data, and once it has captured those correlations, it suggests new candidates with features similar to the top ones seen until that iteration phase (see Fig. 1).

- *The entire approach is data-driven:* This means that the availability of more data, whether from previous attempts to solve the problem or from other state-of-the-art solvers, is expected to enhance performance. In the example of GEO as a booster, we used data explored by Simulated Annealing (SA) but if we had previous observations from any or many other solvers, we could combine it and give it as a starting point to GEO.

- *The model is cost function agnostic, i.e., it is a black box solver.* This is paramount since any cost function can be solved with our approach. Most of the proposals for quantum or quantum inspired optimization require the cost function of the problem to be mapped to a quadratic or polynomial expression. This opens the possibility to tackle any discrete optimization problem, regardless of how complicated or expensive it is to compute the cost function. This is possible since the only information passed to the generative model are the bitstrings who have been explored and their respective cost value.

- *Versatility and Strategic Focus on Portfolio Optimization:* This follows from the item above. The main motivation for selecting the cardinality-constrained portfolio optimization as the NP-hard problem for our study was the availability of concrete benchmarks

and an extensive literature of solvers which have been fine-tuned over the past decades. Every time a new metaheuristic is proposed, chances are portfolio optimization is used to benchmark. Other recent independent works have considered other real-world applications of GEO[8,9]. For example, the authors in ref. 8 considered an industrial case related to a floor planning NP-hard problem. This black-box feature is one of the most prominent ones that render our approach advantageous compared to other quantum heuristics, such as the quantum approximate optimization algorithm (QAOA)[3,4], which relies on the cost function to be a polynomial in terms of the binary variables.

Although other proposals leveraging generative models as a subroutine within the optimizer have appeared recently since the publication of our manuscript (e.g., see GFlowNets[10] and the variational neural annealing[11] algorithms), our framework offers the capability for both: handling arbitrary cost functions (i.e., a blackbox solver) and the possibility to swap the generator for a quantum or quantum-inspired implementation. GEO also has the enhanced feature that the more data is available, the more information can be passed and used to train the (quantum) generator.

As shown in Fig. 1, depending on the GEO specifics we can construct an entire family of solvers whose generative modeling core range from classical, QI or quantum circuit (QC) enhanced, or hybrid quantum-classical model. These options can be realized by utilizing, for example, Boltzmann machines[12] or Generative Adversarial Networks (GAN)[13], Tensor-Network Born Machines (TNBM)[14], Quantum Circuit Born Machines (QCBM)[15] or Quantum-Circuit Associative Adversarial Networks (QC-AAN)[16] respectively,

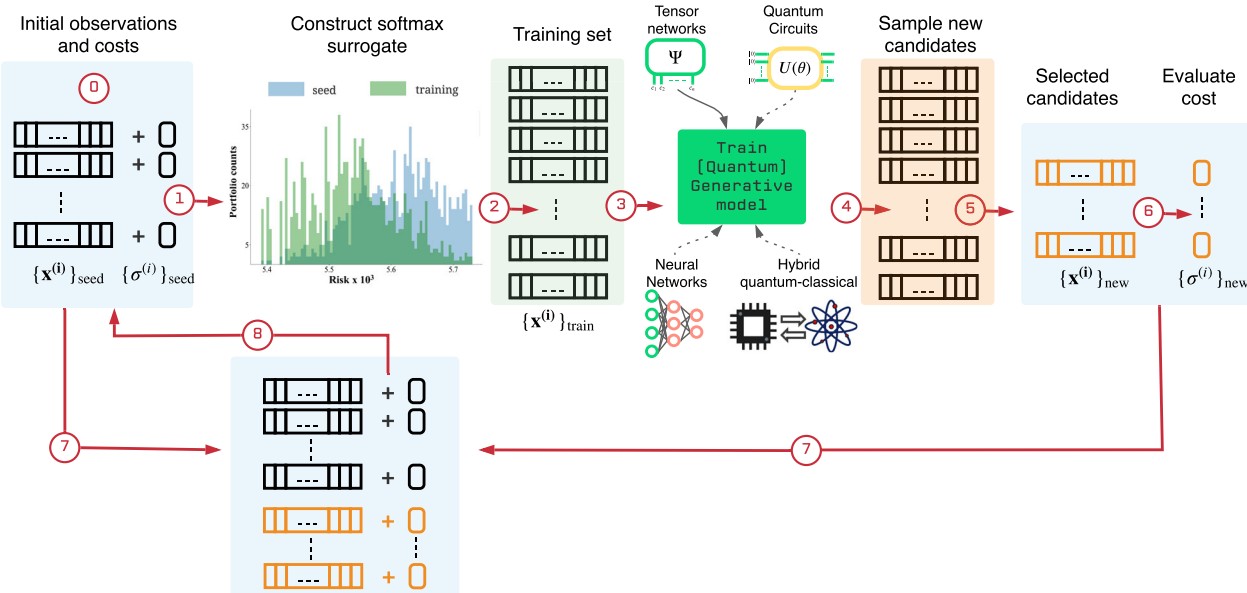

**Fig. 1 | Scheme for our generator-enhanced optimization (GEO) strategy.** The GEO framework leverages generative models to utilize previous samples coming from any quantum or classical solver. The trained quantum or classical generator is responsible for proposing candidate solutions which might be out of reach for conventional solvers. This *seed data set* (step 0) consists of observation bitstrings $\{x^{(i)}\}_{seed}$ and their respective costs $\{\sigma^{(i)}\}_{seed}$. To give more weight to samples with low cost, the seed samples and their costs are used to construct a *softmax* function which serves as a *surrogate* to the cost function but in probabilistic domain. This softmax surrogate also serves as a prior distribution from which the *training set* samples are withdrawn to train the generative model (steps 1–3). As shown in the figure between steps 1 and 2, training samples from the softmax surrogate are biased favoring those with low cost value. For the work presented here, we

implemented a tensor-network (TN)-based generative model. Therefore, we refer to this quantum-inspired instantiation of GEO as TN-GEO. Other families of generative models from classical, quantum, or hybrid quantum-classical can be explored as expounded in the main text. The quantum-inspired generator corresponds to a tensor-network Born machine (TNBM) model which is used to capture the main features in the training data, and to propose new solution candidates which are subsequently post-selected before their costs $\{\sigma^{(i)}\}_{new}$ are evaluated (steps 4-6). The new set is merged with the seed data set (step 7) to form an updated seed data set (step 8) which is to be used in the next iteration of the algorithm. More algorithmic details for the two TN-GEO strategies proposed here, as a *booster* or as a *stand-alone* solver, can be found in the main text and in Supplementary Note 1.F and 1.G.

to name just a few of the many options for this probabilistic component.

QI algorithms come as an interesting alternative since these allow one to simulate larger scale quantum systems with the help of efficient tensor-network (TN) representations. Depending on the complexity of the TN used to build the quantum generative model, one can simulate from thousands of problem variables to a few tens, the latter being the limit of simulating an universal gate-based quantum computing model. This is, one can control the amount of quantum resources available in the quantum generative model by choosing the QI model.

Therefore, from all quantum generative model options, we chose to use a QI generative model based on TNs to test and scale our GEO strategy to instances with a number of variables commensurate with those found in industrial-scale scenarios. We refer to our solver hereafter as *TN-GEO*. For the training of our TN-GEO models we followed the work of Han et al.[17] where they proposed to use Matrix Product States (MPS) to build the unsupervised generative model. The latter extends the scope from early successes of quantum-inspired models in the context of supervised ML[18–21].

In this work, we will discuss two modes of operation for our family of quantum-enhanced solvers:

- In *TN-GEO as a "booster"* we leverage past observations from classical (or quantum) solvers. To illustrate this mode we use observations from simulated annealing (SA) runs. Results are presented in this section and simulation details are provided in Supplementary Note 1.F.
- In *TN-GEO as a stand-alone solver* all initial cost function evaluations are decided entirely by the quantum-inspired generative model, and a random prior is constructed just to give support to the target probability distribution the MPS model is aiming to capture. Results are presented in this section and Simulation details are provided in Supplementary Note 1.G.

Both of these strategies are captured in the algorithm workflow diagram in Fig. 1 and described in more detail in Supplementary Note 1.

To illustrate the implementation for both of these settings, we tested their performance on an NP-hard version of the portfolio optimization problem with cardinality constraints. The selection of optimal investment on a specific set of assets, or *portfolios*, is a problem of great interest in the area of quantitative finance. This problem is of practical importance for investors, whose objective is to allocate capital optimally among assets while respecting some investment restrictions. The goal of this optimization task, introduced by Markowitz[22], is to generate a set of portfolios that offers either the highest expected return (profit) for a defined level of risk or the lowest risk for a given level of expected return. In this work, we focus in two variants of this cardinality constrained optimization problem. The first scenario aims to choose portfolios which minimize the volatility or risk given a specific target return (more details are provided in Supplementary Note 1.A.) To compare with the reported results from the best performing SOTA algorithms, we ran TN-GEO in a second scenario where the goal is to choose the best portfolio given a fixed level of *risk aversion*. This is the most commonly used version of this optimization problem when it comes to comparison among SOTA solvers in the literature (more details are provided in Supplementary Note 1.B.).

The following results are broken into three subsections, each highlighting different features from GEO. First, we focus on GEO as a booster and how it can build from results obtained with other solvers. Second, we focus on GEO as a standalone and compare its performance to SA and the Bayesian optimization library GPyOpt[23]. In the final subsection, we focus on a bencmark comparison of GEO with state-of-the-art solvers. While in TN-GEO as a booster and as a stand-alone solver we implemented and fine-tuned each solver, and in the final benchmark, we leveraged the state-of-the-art results from nine other solvers reported in the last two decades. In the latter case, each

non-GEO solver was thoroughly fine-tuned by the researchers of each reference. This portfolio optimization problem is so canonical that when a new solver is proposed, researchers can compare their results by taking the results from the new proposed solver, as long as the benchmark problems are run in identical conditions. This was one of the main motivations for us to choose this well-established benchmark problem. The "rules of the game" for reporting each market index and performance indicator are reported in Supplementary Note 1.B. In contrast, for the other two subsections, the criteria of evaluation are different, and it emphasizes the performance of GEO when one imposes a limit on the total wall-clock time (e.g., as in the booster mode subsection) and when there is a limited number of calls to the cost function (e.g., as in the stand-alone subsection). The latter is a potential scenario when the bottleneck or expensive step is the cost function evaluation itself (e.g., as it is the case of drug discovery where each evaluation (each candidate molecule) might require synthesis in the lab and an expensive and long process towards its Food and Drug Administration (FDA) approval). Note the desirable condition of the cost function being expensive is only ideal to offset the typical longer time incurred in the steps training the generative model. As shown in teh following subsection, this condition can be significantly relaxed when we use GEO as a booster, since a hybrid strategy where GEO is initialized with previous solutions from other solvers can yield an advantage as well for GEO, even in scenarios where the evaluation of the cost function is inexpensive.

## TN-GEO as a booster for any other combinatorial optimization solver

In Fig. 2 we present the experimental design and the results obtained from using TN-GEO as a booster. In these experiments we illustrate how using intermediate results from simulated annealing (SA) can be used as seed data for our TN-GEO algorithm. As described in Fig. 2A, there are two strategies we explored (strategies 1 and 2) to compare with our TN-GEO strategy (strategy 4). To fairly compare each strategy, we provide each with approximately the same computational wall-clock time. For strategy 2, this translates into performing additional restarts of SA with the time allotted for TN-GEO. In the case of strategy 1, where we explored different settings for SA from the start compared to those used in strategy 2, this amounts to using the same total number of number of cost functions evaluations as those allocated to SA in strategy 2. For our experiments this number was set to 20,000 cost function evaluations for strategies 1 and 2. In strategy 4, the TN-GEO was initialized with a prior consisting of the best 1,000 observations out of the first 10,000 coming from strategy 2 (see Supplementary Note 1.F for details). To evaluate the performance enhancement obtained from the TN-GEO strategy we compute the *relative TN-GEO enhancement*, $\eta$, which we define as

$$\eta = \frac{C_{\min}^{\text{cl}} - C_{\min}^{\text{TN-GEO}}}{C_{\min}^{\text{cl}}} \times 100\%. \tag{1}$$

Here, $C_{\min}^{\text{cl}}$ is the lowest minimum value found by the classical strategy (e.g., strategies 1–3) while $C_{\min}^{\text{TN-GEO}}$ corresponds to the lowest value found with the quantum-enhanced approach (e.g., with TN-GEO). Therefore, positive values reflect an improvement over the classical-only approaches, while negative values indicate cases where the classical solvers outperform the quantum-enhanced proposal.

As shown in the Fig. 2B, we observe that TN-GEO outperforms on average both of the classical-only strategies implemented. The quantum-inspired enhancement observed here, as well as the trend for a larger enhancement as the number of variables (assets) becomes larger, is confirmed in many other investment universes with a number of variables ranging from $N = 30$ to $N = 100$ (see Supplementary Note 3 for more details). Although we show an enhancement compared to SA,

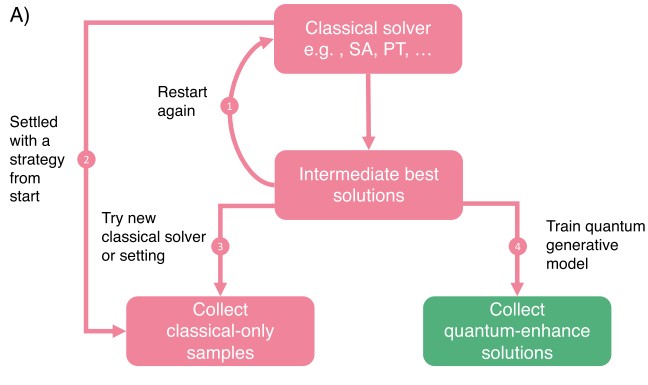

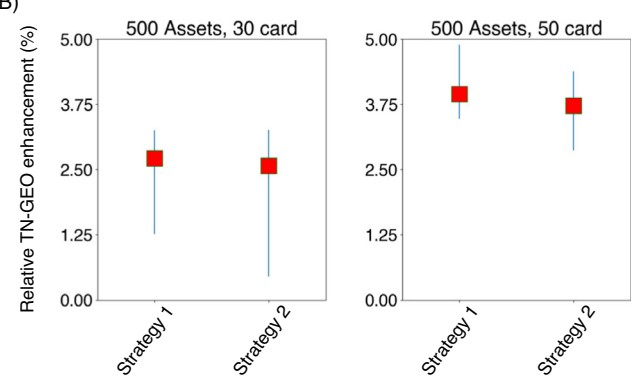

**Fig. 2 | TN-GEO as a _booster_. A** Shows the schematic representation, strategies 1–3 correspond to the current options a user might explore when solving a combinatorial optimization problem with a suite of classical optimizers such as simulated annealing (SA), parallel tempering (PT), generic algorithms (GA), among others. In strategy 1, the user would use its computational budget with a preferred solver. In strategy 2-4 the user would inspect intermediate results and decide whether to keep trying with the same solver (strategy 2), try a new solver or a new setting of the same solver used to obtain the intermediate results (strategy 3), or, as proposed here, to use the acquired data to train a quantum or quantum-inspired generative model within a GEO framework such as TN-GEO (strategy 4). **B** The results show the relative TN-GEO enhancement from TN-GEO over either strategy 1 or strategy 2. Positive values indicate runs where TN-GEO outperformed the respective classical strategies (see Eq. (1)). The data represents bootstrapped medians from 20 independent runs of the experiments and error bars correspond to the 95% confidence intervals. The two instances presented here correspond to portfolio optimization instances where all the assets in the S&P 500 market index where included ($N = 500$), under two different cardinality constraints ($\kappa = 30$ and $\kappa = 50$). This cardinality constraint indicate the number of assets that can be included at a time in valid portfolios, yielding a search space of $M = \binom{N}{\kappa}$, with $M \sim 10^{69}$ portfolios candidates for $\kappa = 50$.

similar results could be expected when other solvers are used, since our approach builds on solutions found by the solver and does not compete with it from the start of the search. Furthermore, the more data available, the better the expected performance of TN-GEO is. An important highlight of TN-GEO as a booster is that these previous observations can come from a combination of solvers, as different as purely quantum or classical, or hybrid.

The observed performance enhancement compared with the classical-only strategy must be coming from a better exploration of the relevant search space, i.e., the space of those bitstring configurations $x$ representing portfolios which could yield a low risk value for a specified expected investment return. That is the intuition behind the construction of TN-GEO. The goal of the generative model is to capture the important correlations in the previously observed data, and to use its generative capabilities to propose similar new candidates.

Generating new candidates is by no means a trivial task in ML and it determines the usefulness and power of the model since it measure its _generalization_ capabilities. In this setting of QI generative models, one expects that the MPS-based generative model at the core of TN-GEO is not simply memorizing the observations given as part of the training set, but that it will provide new unseen candidates. This is an idea which has been recently tested and demonstrated to some extent on synthetic data sets (see e.g., refs. 24–26). In Fig. 3 we demonstrate that our quantum-inspired generative model is generalizing to new samples and that these add real value to the optimization search. This demonstrates the generalization capabilities of quantum generative models in the context of a real-world application in an industrial scale setting, and corresponds to one of the main findings in our paper.

Note that our TN-based generative model not only produces better minima than the classical seed data, but it also generates a rich amount of samples in the low cost spectrum. This bias is imprinted in the design of our TN-GEO and it is the purpose of the _softmax_ surrogate prior distribution shown in Fig. 1. This richness of new samples could be useful not only for the next iteration of the algorithm, but they may also be readily of value to the user solving the application. In some applications there is value as well in having information about the runners-up. Ultimately, the cost function is just a model of the system guiding the search, and the lowest cost does not translate to the best performance in the real-life investment strategy.

### Generator-enhanced optimization as a stand-alone solver

Next, we explore the performance of our TN-GEO framework as a stand-alone solver. The focus is in combinatorial problems whose cost functions are expensive to evaluate and where finding the best minimum within the least number of calls to this function is desired. In Fig. 4 we present the comparison against four different classical optimization strategies. As the first solver, we use the _random_ solver, which corresponds to a fully random search strategy over the $2^N$ bitstrings of all possible portfolios, where $N$ is the number of assets in our investment universe. As second solver, we use the _conditioned random_ solver, which is a more sophisticated random strategy compared to the fully random search. The conditioned random strategy uses the a priori information that the search is restricted to bitstrings containing a fixed number of $\kappa$ assets. Therefore the number of combinatorial possibilities is $M = \binom{N}{\kappa}$, which is significantly less than $2^N$. As expected, when this information is not used the performance of the random solver over the entire $2^N$ search space is worse. The other two competing strategies considered here are SA and the Bayesian optimization library GPyOpt[23]. In both of these classical solvers, we adapted their search strategy to impose this cardinality constraint with fixed $\kappa$ as well (details in Supplementary Note 1.E). This raises the bar even higher for TN-GEO which is not using that a priori information to boost its performance. Specific adaptions of the MPS generative model could be implemented to conserve the number of assets by construction, borrowing ideas from condensed matter physics where one can impose MPS conservation in the number of particles in the quantum state. As explained in Supplementary Note 1.G, we only use this information indirectly during the construction of the artificial seed data set which initializes the algorithm (step 0, Fig. 1), but it is not a strong constraint during the construction of the QI generative model (step 3, Fig. 1) or imposed to generate the new candidate samples coming from it (step 4, Fig. 1). Post selection can be applied _a posteriori_ such that only samples with the right cardinality are considered as valid candidates towards the selected set (step 5, Fig. 1).

In Fig. 4 we demonstrate the advantage of our TN-GEO stand-alone strategy compared to any of these widely-used solvers. In particular, it is interesting to note that the gap between TN-GEO and the other solvers seems to be larger for larger number of variables.

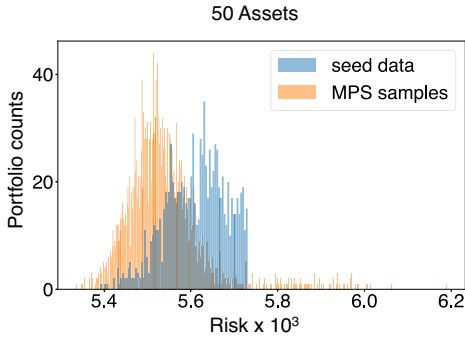
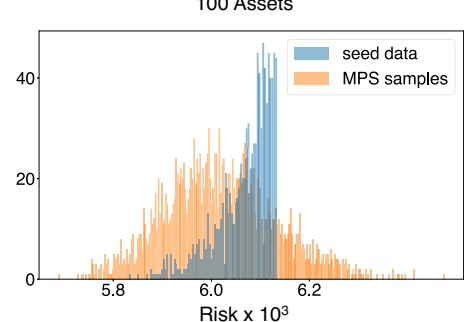

**Fig. 3 | Generalization capabilities of our quantum-inspired generative model.** The blue histogram represents the number of observations or portfolios obtained from the classical solver (*seed data set*), corresponding to an investment universe with $N = 50$ and $N = 100$ assets. In orange we represent samples coming from our quantum generative model at the core of TN-GEO. The green dash line is positioned at the best risk value found in the seed data. This mark emphasizes all the new samples obtained with the quantum generative model and which correspond to lower portfolio risk value (better minima) than those available from the classical solver by itself. The number of samples in the case of $N = 50$ is equal to 31, while 349 samples were obtained from the MPS generative model in the case of $N = 100$.

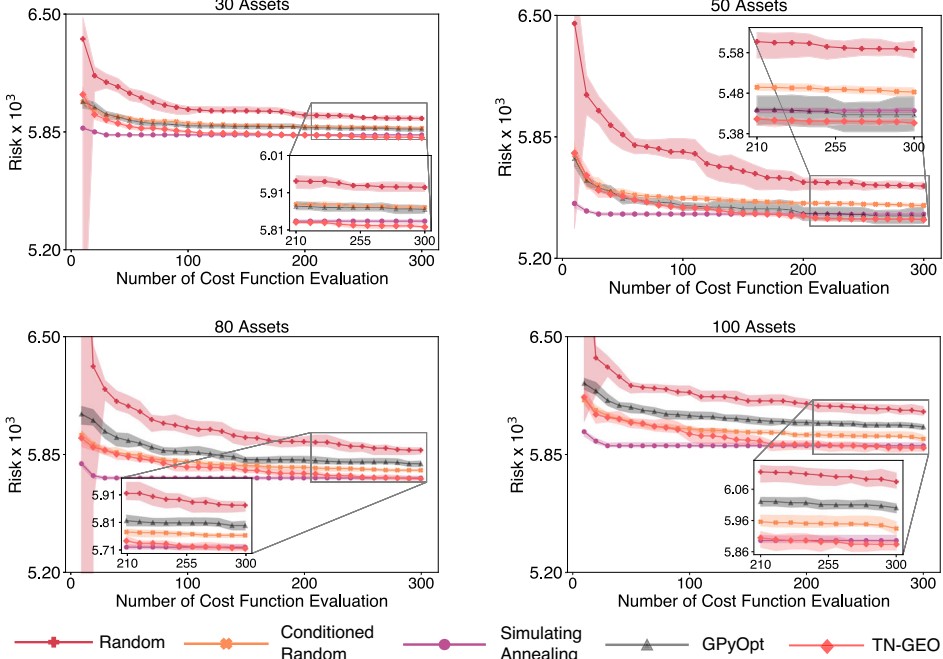

**Fig. 4 | TN-GEO as a *stand-alone* solver.** In this comparison of TN-GEO against four classical competing strategies, investment universes are constructed from subsets of the S&P 500 with a diversity in the number of assets (problem variables) ranging from $N = 30$ to $N = 100$. The goal is to minimize the risk given an expected return which is one of the specifications in the combinatorial problem addressed here. Error bars and their 95% confidence intervals are calculated from bootstrapping over 100 independent random initializations for each solver on each problem. The main line for each solver corresponds to the bootstrapped median over these 100 repetitions, demonstrating the superior performance of TN-GEO over the classical solvers considered here. As specified in the text, with the exception of TN-GEO, the classical solvers use to their advantage the a priori information coming from the cardinality constraint imposed in the selection of valid portfolios.

## Comparison with state-of-the-art algorithms

Finally, we compare TN-GEO with nine different leading SOTA optimizers covering a broad spectrum of algorithmic strategies for this specific combinatorial problem, based on and referred hereafter as: (1) GTS[27], the genetic algorithms, tabu search, and simulated annealing; (2) IPSO[28], an improved particle swarm optimization algorithm[28]; (3) IPSO-SA[29], a hybrid algorithm combining particle swarm optimization and simulated annealing; (4) PBILD[30], a population-based incremental learning and differential evolution algorithm; (5) GRASP[31], a greedy randomized adaptive solution procedure; (6) ABCFEIT[32], an artificial bee colony algorithm with feasibility enforcement and infeasibility toleration procedures; (7) AAG[33], a hybrid algorithm integrating ant colony optimization, artificial bee colony and genetic algorithms; (8) VNSQP[34], a variable neighborhood search algorithm combined with quadratic programming; and, (9) ABC-HP[35], a rapidly converging artificial bee colony algorithm. (10) Additionally, we included a classical version of GEO, based on the Neural Autoregressive Density Estimation (NADE)[36] model as the generator; We refer to this implementation as NADE-GEO.

The test data used by the vast majority of researchers in the literature who have addressed the problem of cardinality-constrained portfolio optimization come from OR-Library[37], which correspond to the weekly prices between March 1992 and September 1997 of the following indexes: Hang Seng in Hong Kong (31 assets); DAX 100 in Germany (85 assets); FTSE 100 in the United Kingdom (89 assets); S&P 100 in the United States (98 assets); and Nikkei 225 in Japan (225 assets). It is important to note that with the exception of NADE-GEO and TN-GEO, each of these nine solvers has been fine-tuned by the authors in the

respective reference. In each, the authors have reported their best results to succeed in this canonical benchmark problem, and those are the values that we report and compare against the two versions of GEO implemented here. Details for the hyperparameter fine-tuning of TN-GEO and NADE-GEO can be found in Supplementary Note 1.H.

Although the full comparison involves the ten algorithms stated above, in this section, we will concentrate on presenting a reduced version of the full results focusing only the current state-of-the-art optimizers, i.e., excluding the metaheuristics from the early 2000's, and including NADE-GEO. In particular, the selected algorithms whose results are presented in this section are GRASP, ABCFEIT, AAG, VNSQP, ABC-HP and NADE-GEO. Full results are presented in Supplementary Note 3.

Therefore, here we present the results obtained with TN-GEO and its comparison with NADE-GEO and five of the different SOTA metaheuristic algorithms mentioned above whose results are publicly available in the literature. Table 1 shows the results of those algorithms and all performance metrics for each of the five index data sets (for more details on the evaluation metrics, see Supplementary Note 1.B).

Each algorithm corresponds to a different column, with TN-GEO in the rightmost column. The values are shown in *italic* entities if the TN-GEO algorithm performed better or equally well compared to the other algorithms on the corresponding performance metric. The numbers in **bold** mean that the algorithm found the best (lowest) value across all algorithms.

From all the entries in this table, 67% of them correspond to *italic* entries, where TN-GEO either wins or draws, which is a significant percentage giving that these optimizers are among the best reported in the last decades.

In Table 2 we show a pairwise comparison of TN-GEO against each of the six selected SOTA optimizers. This table reports the number of times TN-GEO wins, loses, or draws compared to results reported for the other optimizer, across all the performance metrics and for all the 5 different market indexes. Therefore, we report in the same table the overall percentage of wins plus draws in each case. We see that this percentage is greater than 50% in all the cases.

Furthermore, in Table 2, we use the Wilcoxon signed-rank test [38], which is a widely used nonparametric statistical test used to evaluate

**Table 1 | Detailed comparison with SOTA algorithms for each of the five index data sets and on seven different performance indicators described in Supplementary Note 1.B**

| Data Set | Performance indicator | GRASP[31] | ABCFEIT[32] | AAG[33] | VNSQP[34] | ABC-HP[35] | NADE-GEO | TN-GEO |
|---|---|---|---|---|---|---|---|---|
| Hang Seng | Mean | *1.0965* | 1.0953 | *1.0965* | *1.0964* | **1.0873** | *1.1007* | 1.0958 |
| | Median | 1.2155 | *1.2181* | *1.2181* | 1.2155 | **1.2154** | 1.2170 | 1.2181 |
| | Min | **0.0000** | **0.0000** | **0.0000** | **0.0000** | **0.0000** | **0.0000** | **0.0000** |
| | Max | **1.5538** | **1.5538** | **1.5538** | **1.5538** | **1.5538** | **1.5538** | **1.5538** |
| | MEUCD | **0.0001** | **0.0001** | **0.0001** | **0.0001** | **0.0001** | **0.0001** | **0.0001** |
| | VRE | *1.6400* | 1.6432 | *1.6395* | *1.6397* | **1.6342** | *1.6429* | 1.6392 |
| | MRE | 0.6060 | 0.6047 | 0.6085 | 0.6058 | **0.5964** | 0.6079 | 0.6082 |
| DAX100 | Mean | *2.3126* | *2.3258* | *2.3130* | *2.3125* | **2.2898** | *2.3125* | 2.3142 |
| | Median | 2.5630 | 2.5678 | 2.5587 | 2.5630 | 2.5629 | 2.5630 | 2.5660 |
| | Minimum | *0.0059* | *0.0023* | *0.0023* | *0.0059* | *0.0059* | *0.0059* | **0.0023** |
| | Maximum | **4.0275** | **4.0275** | **4.0275** | **4.0275** | **4.0275** | **4.0275** | **4.0275** |
| | MEUCD | **0.0001** | **0.0001** | **0.0001** | **0.0001** | **0.0001** | **0.0001** | **0.0001** |
| | VRE | *6.7593* | 6.7925 | *6.7806* | *6.7583* | 6.8326 | *6.7591* | **6.7540** |
| | MRE | 1.2769 | 1.2761 | 1.2780 | 1.2767 | **1.2357** | 1.2765 | 1.2763 |
| FTSE100 | Mean | *0.8451* | *0.8481* | *0.8451* | *0.8453* | **0.8406** | *0.8647* | 0.8445 |
| | Median | **1.0841** | **1.0841** | **1.0841** | **1.0841** | **1.0841** | **1.0841** | **1.0841** |
| | Minimum | 0.0016 | 0.0047 | **0.0006** | 0.0045 | 0.0016 | 0.0016 | 0.0047 |
| | Maximum | **2.0576** | 2.0638 | 2.0605 | 2.0669 | 2.0670 | 2.3718 | 2.0775 |
| | MEUCD | **0.0000** | **0.0000** | **0.0000** | **0.0000** | **0.0000** | **0.0000** | **0.0000** |
| | VRE | *2.4350* | 2.4397 | *2.4350* | *2.4349* | **2.4149** | 2.4713 | 2.4342 |
| | MRE | 0.3245 | 0.3255 | **0.3186** | 0.3252 | 0.3207 | 0.3235 | 0.3254 |
| S&P100 | Mean | *1.2937* | *1.2930* | *1.2930* | **1.2649** | *1.3464* | *1.3314* | 1.2918 |
| | Median | 1.1420 | 1.1369 | **1.1323** | **1.1323** | *1.1515* | 1.1420 | 1.1452 |
| | Minimum | *0.0009* | **0.0000** | **0.0000** | **0.0000** | *0.0009* | *0.0009* | **0.0000** |
| | Maximum | 5.4551 | **5.4422** | 5.4642 | 5.4551 | 5.4520 | 6.7448 | 5.4422 |
| | MEUCD | **0.0001** | **0.0001** | **0.0001** | **0.0001** | **0.0001** | **0.0001** | **0.0001** |
| | VRE | 2.5211 | 2.5260 | 2.5255 | **2.5105** | 2.5364 | 2.5975 | 2.5269 |
| | MRE | 0.9063 | 0.8885 | **0.7044** | 0.9072 | 0.8858 | 0.9064 | 0.9117 |
| Nikkei | Mean | 0.5782 | 0.5781 | 0.5781 | *0.5904* | **0.5665** | 1.0391 | 0.5793 |
| | Median | *0.5857* | *0.5856* | 0.5854 | *0.5857* | 0.5858 | *0.6500* | 0.5855 |
| | Minimum | **0.0000** | **0.0000** | **0.0000** | **0.0000** | **0.0000** | 0.2270 | **0.0000** |
| | Maximum | **1.1606** | **1.1606** | 1.1607 | **1.1606** | **1.1606** | 7.4928 | **1.1606** |
| | MEUCD | **0.0000** | **0.0000** | **0.0000** | **0.0000** | **0.0000** | **0.0000** | **0.0000** |
| | VRE | *0.8359* | *0.8396* | **0.8191** | *0.8561* | 0.8314 | 1.6873 | 0.8353 |
| | MRE | 0.4184 | 0.4147 | 0.4233 | 0.4217 | **0.4042** | *0.4256* | 0.4229 |

Entries in *italic* correspond to cases where TN-GEO performed better or tied compared to the other algorithm. Entries in **bold**, corresponding to the best (lowest) value, for each specific indicator.

**Table 2 | Pairwise comparison of TN-GEO against each of the SOTA optimizers**

| TN-GEO vs Other: | GRASP | ABCFEIT | AAG | VNSQP | ABC-HP | NADE-GEO |
|---|---|---|---|---|---|---|
| Wins(+) | 12 | 10 | 11 | 11 | 8 | 8 |
| Loss( − ) | 12 | 9 | 11 | 12 | 16 | 18 |
| Ties | 11 | 16 | 13 | 12 | 11 | 9 |
| (Wins+Ties)/Total | 66% | 74% | 69% | 66% | 54% | 77% |
| Asymptotic significance ($p$) | 0.247 | 0.888 | 0.363 | 0.594 | 0.110 | 0.003 |
| Decision | Retain | Retain | Retain | Retain | Retain | Reject |

The asymptotic significance is part of the Wilcoxon signed-rank test results. The null hypothesis that the performance of the two algorithms is the same is tested at the 95% confidence level (significance level: $\alpha = .05$). Results show that TN-GEO is on par with all the SOTA algorithms. We also report the count for TN-GEO wins, losses, and ties, compared to each of the other algorithms.

and compare the performance of different algorithms in different benchmarks [39]. Therefore, to statistically validate the results, a Wilcoxon signed-rank test is performed to provide a meaningful comparison between the results from TN-GEO algorithm and the selected SOTA metaheuristic algorithms. The Wilcoxon signed-rank test tests the null hypothesis that the median of the differences between the results of the algorithms is equal to 0. Thus, it tests whether there is no significant difference between the performance of the algorithms. The null hypothesis is rejected if the significance value ($p$) is less than the significance level ($\alpha$), which means that one of the algorithms performs better than the other. Otherwise, the hypothesis is retained.

As can be seen from the table, the null hypotheses are accepted at $\alpha = 0.05$ for the TN-GEO algorithm over these recent SOTA algorithms except for NADE-GEO, that is rejected, meaning that TN-GEO performs significantly better than NADE-GEO. Thus, in terms of performance on all metrics combined, the results show that there is no significant difference between TN-GEO and the five selected SOTA optimizers (GRASP, ABCFEIT, AAG, VNSQP, and ABC-HP).

In particular, TN-GEO is on par with the most competitive of all the solvers, referred to here as ABC-HP. The authors of ref. 35 attribute the success of this recent ant bee colony solver to a good balance of diversification (good exploration of the search space) and intensification (search around regions in the neighborhood of local minima). Since GEO generates its candidates from the correlations learned from the data, it is not restricted to local search but can be considered a global search solver, which is a difficult property to include in most of the solvers, which usually only exploit the local neighborhood of the best intermediate solutions. Overall, the results confirm the competitiveness of our quantum-inspired proposed approach against SOTA metaheuristic algorithms. This is remarkable, considering that these metaheuristics have been explored and fine-tuned for decades.

## Discussion

Compared to other quantum optimization strategies, an important feature of TN-GEO is its algorithmic flexibility. As shown here, unlike other proposals, our GEO framework can be applied to arbitrary cost functions, which opens the possibility of new applications that cannot be easily addressed by an explicit mapping to a polynomial unconstrained binary optimization (PUBO) problem. Our approach is also flexible with respect to the source of the seed samples, as they can come from any solver, possibly more efficient or even application-specific optimizers. The demonstrated generalization capabilities of the generative model that forms its core, helps TN-GEO build on the progress of previous experiments with other state-of-the-art solvers, and it provides new candidates that the classical optimizer may not be able to achieve on its own. We are optimistic that this flexible approach will open up the broad applicability of quantum and quantum-inspired

generative models to real-world combinatorial optimization problems at the industrial scale.

Although we have limited the scope of this work to tensor network-based generative quantum models, it would be a natural extension to consider other generative quantum models as well. For example, hybrid classical-quantum models such as quantum circuit associative adversarial networks (QC-AAN) [16] can be readily explored to harness the power of generative quantum models with so-called noisy intermediate-scale quantum (NISQ) devices [40]. In particular, the QC-AAN framework opens up the possibility of working with a larger number of variables and going beyond discrete values e.g., variables with continuous values as those found in Mixed Integer Programming (MIP) optimization problems [41] or [42]. Both quantum-inspired and hybrid quantum-classical algorithms can be tested in this GEO framework in even larger problem sizes of this NP-hard version of the portfolio optimization problem or any other combinatorial optimization problem. As the number of qubits in NISQ devices increases, it would be interesting to explore generative models that can utilize more quantum resources, such as Quantum Circuit Born Machines (QCBM) [15]: a general framework to model arbitrary probability distributions and perform generative modeling tasks with gate-based quantum computers.

The question of whether a significant advantage can be obtained from GEO by using quantum devices is an active research topic currently being explored. One proposal to reach a more systematic and incremental enhancement from the best quantum-inspired solution to an enhanced quantum-hardware realization was recently proposed in ref. 43. There, one starts from the best available quantum-inspired tensor-network solution and maps it to a quantum circuit. This can be subsequently modified by adding gates beyond those from the decomposition to increase the plausible correlations beyond those accessible with the quantum-inspired tensor-network-based solution. The access to longer-range correlations enhances, in turn, the expressibility of the quantum generative model while taking it beyond the capabilities of classical simulation. In that work, the specific case of generative models was illustrated, and therefore, these recent decomposition techniques can be directly applied to extend the capabilities of TN-GEO explored here, and, as the technologies mature and the level of noise is reduced, explore these enhanced models directly on quantum devices. Additionally, in ref. 44 a comparison of quantum generative models with state-of-the-art classical generative models, such as Transformers and Recurrent Neural Networks, was presented, and the results were very encouraging in the data sets studied.

Increasing the expressive power of the quantum-inspired core of MPS to other more complex but still efficient QI approaches, such as tree-tensor networks [45], is another interesting research direction. Although we have fully demonstrated the relevance and scalability of our algorithm for industrial applications by increasing the performance of classical solvers on industrial scale instances (all 500 assets in the S&P 500 market index), there is a need to explore the performance improvement that could be achieved by more complex TN representations or on other combinatorial problems.

Although the goal of GEO was to show good behavior as a general black-box algorithm without considering the specifics of the study application, it is a worthwhile avenue to exploit the specifics of the problem formulation to improve its performance and runtime. In particular, for the portfolio optimization problem with a cardinality constraint, it is useful to incorporate this constraint as a natural MPS symmetry, thereby reducing the effective search space of feasible solutions from the size of the universe to the cardinality size. While imposing such constraints is possible with tensor-networks constructions as recently demonstrated in ref. 46, there does not seem to be a native way to add such common constraints in canonical deep learning models based on neural-network units.

Beyond the strategy of GEO as a booster, another way to use GEO in conjunction with classical solvers is to use it as the optimization subroutine for the smaller subproblems originating from decomposition or multilevel techniques [see for e.g., refs. 47–49] used to mitigate the limitation in the number of qubits in NISQ devices.

Usually, these subproblems are solved with gate-based quantum optimization heuristics such as the Quantum Approximate Optimization Algorithm (QAOA)[3] or D-wave annealing devices, but one could implement a quantum-circuit version of GEO, for example, using QCBM as the generative models, to solve these smaller subproblems and assist the solution of the larger problem via the hybrid quantum-classical decomposition approach. The general question of the hardware requirements needed to prove quantum advantage is a challenging one, and it is beyond the scope of this work, but we think GEO opens the possibility to start exploring this question in more realistic scenarios and in more general cost functions than those that can be natively considered with other approaches such as QAOA.

Finally, our thorough comparison with SOTA algorithms, which have been fine-tuned for decades on this specific application, shows that our TN-GEO strategy manages to outperform a couple of these and is on par with the other seven optimizers. This is a remarkable feat for this approach and hints at the possibility of finding commercial value in these quantum-inspired strategies in large-scale real-world problems, as the instances considered in this work. Also, it calls for more fundamental insights towards understanding when and where it would be beneficial to use this TN-GEO framework, which relies heavily on its quantum-inspired generative ML model. For example, understanding the intrinsic bias in these models, responsible for their remarkable performance, is another important milestone on the road to practical quantum advantage with quantum devices in the near future. The latter can be asserted given the tight connection of these quantum-inspired TN models to fully quantum models deployed on quantum hardware. And this question of when to go with quantum-inspired or fully quantum models is a challenging one that we are exploring in ongoing future work.

## Data availability
The data generated in this study is available at: https://doi.org/10.5281/zenodo.10668479

## Code availability
The code used to generate the data in this study has been deposited at: https://doi.org/10.5281/zenodo.10668479. Your access to and use of the downloadable code (the "Code") contained in this Section is subject to a non-exclusive, revocable, non-transferable, and limited right to use the Code for the exclusive purpose of undertaking academic, governmental, or not-for-profit research. Use of the Code or any part thereof for commercial or clinical purposes is strictly prohibited in the absence of a Commercial License Agreement from Zapata AI.

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

## Acknowledgements

We would like to acknowledge Manuel S. Rudolph, Marta Mauri, Matthew J.S. Beach, Yudong Cao, Luis Serrano, Jhonathan Romero-Fontalvo, Brian Dellabetta, Matthew Kowalsky, Jacob Miller, John Realpe-Gomez, and Collin Farquhar for their feedback on an early version of this manuscript

## Author contributions

A.P.-O. conceived the idea of the GEO framework. J.A., M.G.V., and A.P.-O. contributed to its final form. J.A. performed all the numerical experiments and results for GEO as a booster and as a standalone solver. M.G.V., J.A., and C.B.K. contributed to the results section comparing GEO with state-of-the-art solvers and the respective statistical analysis. M.G.V. implemented the NADE-GEO solver. A.P.-O. helped supervise and coordinate the efforts in this work. All authors regularly analyzed the numerical results and contributed to the final version of the manuscript.

## Competing interests

The authors declare the following competing interests: J.A., M.G.V., C.B.K., and A.P.-O. were employed by Zapata Computing Canada Inc. during the development of this work. There was no collaboration between Acadian Asset Management LLC. and Zapata AI during the development of this work.

## Additional information

**Peer review information** : *Nature Communications* thanks the anonymous reviewer(s) for their contribution to the peer review of this work. A peer review file is available.

