## [Peer Review File · Nature Communications]

Enhancing Combinatorial Optimization with Classical and Quantum Generative ModelsREVIEWER COMMENTS

Reviewer #1 (Remarks to the Author):

The authors propose a framework that incorporates quantum or classical generative models to tackle optimization problems.

The main focus of the work is problems related to portfolio optimization and a quantum-inspired instantiation of the framework.

The paper is generally well-written and presents an interesting approach and results. However in its present form I have difficulty assessing the significance, scope, and impact of the results, which are paramount to my recommendation.

Hence at present I recommend additional revision of the manuscript.

I provide some further comments below.

Comments:

-What is scope and impact of the results? A general framework is described, but what specific instantiations and cases should we expect to provide greatest advantage? (I clearly see, at minimum, a classical-ML approach to portfolio optimization, and would appreciate more clarity regarding more general settings)

-Results are shown for portfolio optimization, but wasn't clear to what other important problems the approach is best suited for or should be expected to be advantageous.

-In particular, how common are problems where "cost function evaluation can be very expensive" ?

Can this be quantified? For instance, what is the resource tradeoff between query cost and runtime? Is your approach still competitive if this is relaxed?

-The results rely on a quantum inspired method, what evidence or support do we have that

using actual quantum devices will improve further? How should we expect GEO to perform for other quantum or classical models?

-For the quantum case, is the proposed method not still severely limited by the underlying quantum hardware and quantum model resource requirements? (in contrast, for instance, to problem decomposition approaches where the goal may be to accommodate fewer quantum resources)

-Regarding "GTS" optimizer in Section III.C and the subsequent reported results for it, I was unclear what was meant. Each of G,T,S refers to an independent optimization strategy, are you running each and reporting the best?

-Similarly, I was left with follow up questions such as to what degree were these (G,T,S) as well as the other nine "leading SOTA optimizers" tailored/optimized for the problem at hand? (In regards to truly 'fair' comparison in the reported numbers)

-Table I is not so easy to read with the many entries. Is it worth including first 4 optimizers here? They don't seem competitive (no to mention their columns are mostly "-")s)

-RCABC appeared to perform comparably to TN-GEO, it would be nice to see more discussion of this or even a more detailed comparison

Reviewer #2 (Remarks to the Author):

**** Key results ****

This paper proposes to use a quantum inspired generative model to help more efficiently explore the space of feasible solutions for combinatorial optimization problems. The generative model is used to sample new candidate solutions: from a set of already explored solutions with their associated cost, the generative model is trained to learn a distribution over the solution space for which the probability of each seen solution is proportional to their associated cost, thus potentially making it possible to sample new promising candidate

solutions. The authors focus on using a quantum inspired generative model based on tensor networks (matrix product states), which have been previously introduced in [15].

The proposed approach is evaluated on the task of portfolio optimization and compared with state of the art optimization algorithms for this task. The experiments reveal that the approach is competitive with these state of the art solvers (which have been fine tuned for decades), even outperforming some of them on this particular task.

**** Validity ****

The results presented in the paper are valid. Both the methodology and the experiment setup is sound.

**** Data & methodology ****

The paper is overall clearly written and the experiments demonstrate well the potential benefits and usefulness of the method. One concern that I have is that the advantage of using a quantum approach is not clearly demonstrated. In particular, it is not clear how the fact that quantum inspired models were chosen for the generative part is key to obtaining the experimental results. At the very least I believe the proposed approach should be compared with replacing the MPS model with a simple HMM learned using the Baum-Welch (i.e. EM) algorithm. Other generative models should be considered as baselines as well (e.g. a simple RNN trained using backpropagation through time or a more complex model such as NADE [Uria et al]). It may be the case that replacing the MPS model by such an alternative non-quantum inspired model would lead to similar result. To sum up, I believe the authors should experimentally investigate and discuss more in depth to which extent the quantum part of their approach is necessary and beneficial.

Uria, Benigno, et al. "Neural autoregressive distribution estimation." *The Journal of Machine Learning Research* 17.1 (2016): 7184-7220.

**** Appropriate use of statistics and treatment of uncertainties ****

Yes, the results are reported appropriately using classical statistical tools and treatment of uncertainties.

**** Conclusions: Do you find that the conclusions and data interpretation are robust, valid and reliable? ****

Overall the conclusions and interpretation of the experimental results are robust and reliable. Though, as I mentioned above, I believe there is a set of experiments missing which would demonstrate more the relevance of a quantum approach to the generative model part of the proposed method.

**** References ****

The manuscript references previous literature appropriately.

**** Clarity and context ****

The paper is very well written and structured, and easy to follow for someone familiar with tensor network models.

**** Suggested improvements ****

- Include an experiment to compare the proposed approach with non-quantum generative models
- It could be a nice addition to have experiments on another combinatorial optimization problem than portfolio optimization.
- Related to the previous point, the authors may consider presenting their approach in a more general context rather than specifically for the portfolio optimization problem. As I understood it, the proposed approach can be applied to many different kind of optimization

problems but the current presentation can suggest that the method is tailored specifically for portfolio optimization. A more general presentation of the method, as well as a clear explanation of the different kind of problems the approach can be applied to, could make for a more impactful paper (by reaching a wider audience). For example, can the approach be applied to any MIP?

Reply to Reviewer #1:

The authors propose a framework that incorporates quantum or classical generative models to tackle optimization problems. The main focus of the work is problems related to portfolio optimization and a quantum-inspired instantiation of the framework.

The paper is generally well-written and presents an interesting approach and results. However in its present form I have difficulty assessing the significance, scope, and impact of the results, which are paramount to my recommendation.

Hence at present I recommend additional revision of the manuscript. I provide some further comments below.

Reply: We thank the reviewer the careful revision our manuscript, and for the positive feedback about the current form of the manuscript. We address each of the questions below with the hope this helps to clarify the scope and significance of our work.

-What is scope and impact of the results? A general framework is described, but what specific instantiations and cases should we expect to provide greatest advantage?(I clearly see, at minimum, a classical-ML approach to portfolio optimization, and would appreciate more clarity regarding more general settings)

-Results are shown for portfolio optimization, but wasn't clear to what other important problems the approach is best suited for or should be expected to be advantageous.

Reply: We would like to mention that beyond our proposal being a “a classical-ML approach to portfolio optimization”, the baseline of our contribution is a ML framework that leverages classical, quantum-inspired, or quantum generative models for solving **any** combinatorial problems. Although not a requirement for using the framework, having the possibility to easily generate bitstrings in the valid solution space (as the case studied here, i.e., bitstrings with a specified cardinality) can be useful since these synthetic data points can be passed as a warm start for the generative model. This in turn can help guide the trained generative model towards the right support of the probability distribution to be explored.

Using ML techniques for combinatorial problems is not a straightforward and rather a new research domain. More specifically, although other reinforcement learning and deep learning approaches had been proposed (see e.g., Ref. [1] and [2]), to the best of our knowledge our work is the first proposal using generative models for combinatorial optimization tasks. This is also the first proposal **flexible** enough to easily incorporate and explore quantum-inspired or quantum generative models. At first, we thought of adapting or enhancing existing reinforcement learning and deep learning proposals with quantum models, but this was not practical or at least not obvious how to do it. Instead, we noticed using generative models could be a very promising approach, given its prominence as candidates for practical quantum advantage. But this required to go beyond the incremental adaptation and to design the algorithm in a completely different framework, which resulted in the GEO proposal here.

Here are more general settings that make this approach advantageous over other available solvers:

- 1) **The entire approach is data-driven:** what this implies is that the more data is available, either from previous attempts to solve the problem with other state-of-the-art solvers, the better the performance is expected. In the example of GEO as a booster we used data explored by Simulated Annealing (SA) but if we had previous observations from any or many other solvers, we could combine it and give it as a starting point to GEO.
- 2) **It leverages the power of generative models:** The essence of the solver is that it is aiming to unveil non-obvious structure in the data, and once it has captured those correlations, it suggests new outstanding candidates with features similar to the top ones seen until that iteration phase.
- 3) **the model is cost function agnostic, i.e., it is a black-box solver.** This is paramount since any cost function can be solved with our approach. Most of the proposals for quantum or quantum inspired optimization require the cost function of the problem to be mapped to a quadratic or polynomial expression. This black-box feature open the possibility to tackle any discrete optimization problem, regardless of how complicated or expensive it is to compute the cost function. This is possible within GEO since the only information passed to the generative model are the bitstrings who have been explored and their respective cost value.
- 4) **Nothing is special about the portfolio optimization problem.** This follows from the item above. The main motivation for selecting this specific instantiation of portfolio optimization was the availability of concrete benchmarks and an extensive literature of solvers which have been fine-tuned for over the past decades. Every time a new metaheuristic is proposed, chances are this cardinality-constrained portfolio optimization problem is used to benchmark. Other recent independent works have considered other real-world applications of GEO. For example, in Ref. [3], the authors considered an industrial case related to a floor planning NP-hard problem. This black-box feature is one of the most prominent ones which render our approach advantageous compared to other quantum heuristics, such as the quantum approximate optimization algorithm (QAOA), which relies on the cost function to be a polynomial in terms of the binary variables.

Although the items above only point to features which make the approach valuable and different when compared to optimizers proposed to date, it is still to be explored if similar advantages can be observed in other family of important problems. Although this question is an active research question, there has been some progress in understanding which datasets or problems might benefit the most. In a recent publication, (see Ref. [4]), we showed how the search space of problems with equality constrains, can be represented efficiently in quantum-inspired generative models, but this is not the case for deep-neural-network-based generative models. Besides the demonstration presented here pointing that quantum and quantum-inspired models are a promising route to be explored, this more recent paper poses a more concrete differentiator between the capabilities of quantum-inspired models with respect to traditional deep learning models. These equality constraints appear in a broad family of problems, including the cardinality constrained portfolio optimization studied here, but also it extends to problems with any linear equality constrains.

Revisions to the manuscript: The new Introduction has been significantly rewritten and restructured for clarity and to indicate more explicitly these salient highlights indicated above. We also revised the Outlook section adding some material related to the recent developments towards the identification of data sets which might benefit from quantum-inspired and quantum models over state-of-the-art generative models

-In particular, how common are problems where "cost function evaluation can be very expensive" ? Can this be quantified? For instance, what is the resource tradeoff between query cost and runtime? Is your approach still competitive if this is relaxed?

Reply: We thank the referee for bringing this point since some clarifications can be added to the manuscript. Note that this desirable condition of having an expensive cost function applies mainly to *GEO as a standalone*, although it is important to note that even in that setting it is not a necessary condition. The reason is because the overhead of training the generative model might have a larger impact for that setting than in the booster mode, where the initial available data can be seen as a warm start. It can be noted that even in problems with cost function which might be quick to evaluate, as long as the problem is hard and a solution is not reached with a competitor solver, it might be still worth to use GEO as a standalone which might yield a completely different set of solutions, or even better ones, since its search strategy is completely different.

This condition of an expensive cost function was relaxed altogether in the setting of GEO as a *booster* (described in Sec. II A) where the comparison criteria was moved from number of cost function evaluations to runtime. In that study, we used a similar criteria as that suggested by the referee, where we used the trade off between query cost and runtime by using the more generic and practical criteria of fixed total wall clock runtime used for each of the algorithmic strategies. There, the time it takes to evaluate the cost function and the training of the generative models are considered, since these are part of the total time it takes to run each algorithm. In that comparison, between Simulated Annealing (SA) and GEO, although the training of the generative model within GEO takes longer, compared to the quick updates in SA, we still see an advantage in adopting GEO as a booster, which consisted of GEO being initialized with partial solutions obtained from SA. We show in that section that it is more efficient to change the strategy to GEO, than to insist and continue performing runs just with SA, if both algorithmic strategies are given the same total wallclock time.

Revisions to the manuscript: We have added further clarification in the new paragraph right before subsection IIA.

-The results rely on a quantum inspired method, what evidence or support do we have that using actual quantum devices will improve further? How should we expect GEO to perform for other quantum or classical models?

Reply: The question of whether a significant advantage can be obtained by using quantum devices is an active research topic and certainly needs to be explored further. One proposal to reach a more systematic and incremental enhancement from the best quantum-inspired solution to an enhanced quantum-hardware realization was recently proposed in Ref. [5]. There, one starts from the best available quantum-inspired tensor-network solution and maps it to a quantum circuit. This can be subsequently modified by adding gates beyond those from the decomposition to increase the plausible correlations beyond those accessible with the quantum-inspired tensor-network-based solution. The access to longer-range correlations enhances, in turn, the expressibility of the quantum generative model while taking it beyond the capabilities of classical simulation. In that work, the specific case of generative models was illustrated, and therefore, these novel decomposition techniques can be directly applied to extend the capabilities of TN-GEO explored here, and, as the technologies mature and the level of noise is reduced, explore these enhanced models directly on quantum devices. Additionally, in Ref. [6] a comparison of quantum

generative models with state-of-the-art classical generative models was presented, and the results were very encouraging in the data sets studied.

To the comment of the performance of TN-GEO against other classical models, and as part of the request from the second referee, we added a classical version of GEO, based on the Neural Autoregressive Density Estimation (NADE) model [7]. In this revised version of the manuscript, we showed that although NADE-GEO is competitive with the early solvers (from about a decade ago), it is underperforming compared to TN-GEO and the other state-of-the-art solvers within the past five years. Although we don't expect the results to be universal, in this specific TN-GEO versus NADE-GEO comparison we still see an advantage for the quantum-inspired over the classical generative neural network model, with the advantage as well that the TN-base model has less hyperparameters to fine tune. Each of these comparisons is an extensive amount of work, and we feel that despite being beyond the scope of this particular work, our work opens the possibility to explore quantum and quantum-inspired generative models toward solving arbitrary combinatorial optimization, and as a concrete framework to study quantitatively practical quantum advantage with future quantum technologies (see e.g., Ref. [6] whose framework is inspired in this work)

Revisions to the manuscript: We complemented the Outlook where we had already mentioned about the opportunities and challenges that arise from thinking of an implementation in quantum devices. In particular, we added some of the discussion above around the new references that appeared after we submitted our work. We have also expanded the Results & Discussion subsection, Sec. IIC, with the respective discussion related to NADE-GEO and TN-GEO.

-For the quantum case, is the proposed method not still severely limited by the underlying quantum hardware and quantum model resource requirements? (in contrast, for instance, to problem decomposition approaches where the goal may be to accommodate fewer quantum resources)

Reply: If we correctly understand the proposal from the Reviewer, the reviewer is thinking of problem decomposition techniques which divide the problem into many smaller subproblems to accommodate the number of qubits in near-term technologies. From a practical point of view, for these techniques to be promising for quantum technologies, one still needs the subproblems still be intractable or hard to solve by conventional classical solvers, otherwise, it would not be worth to submit them to a quantum hardware. If the instances happen to be indeed intractable, then GEO can still be used to solve these subproblems, since in the cases of decomposition techniques we are familiar with, it is still required that solutions to the subproblem needs to be gathered before they are combined or used to solve the larger problem (e. g., see Refs. [8], [9], and [10] below and that have been added to the revised manuscript). In other words, we don't see it necessarily as one strategy versus the other, since we think of GEO as a standalone strategy to solving hard optimization problems or subproblems. It is important to note that, with now gate-based devices reaching the level of hundreds of qubits, such techniques and demonstrations can become more relevant. But we can see how GEO can be incorporated as well in the solution of the smaller but intractable partitions which are sent to these quantum devices.

To the question of the limitation of quantum hardware resources, not only in terms of number of qubits, but also in terms of connectivity, this is one is an important one and we will be addressing that in a theoretical and experimental ongoing work in our team. In that study, noise is also factored as part of the experimental demonstration.

Revisions to the manuscript: We have complemented the Outlook addressing both, the potential interplay with other decomposition techniques, and the open questions and challenges related to hardware limitations.

-Regarding "GTS" optimizer in Section III.C and the subsequent reported results for it, I was unclear what was meant. Each of G,T,S refers to an independent optimization strategy, are you running each and reporting the best?

-Similarly, I was left with follow up questions such as to what degree were these (G,T,S) as well as the other nine "leading SOTA optimizers" tailored/optimized for the problem at hand? (In regards to truly 'fair' comparison in the reported numbers)

Reply: It is important to note that we have not fine-tuned all the solvers ourselves. The Results section of the papers is broken into three subsections, each highlighting different features from GEO. Sec. IIA focuses on GEO as a booster and how it can build from results obtained with other solvers. Sec. IIB focuses on GEO as a standalone and compares its performance to SA and the Bayesian optimization library GPyOpt. Finally, Sec. IIC focuses on a comparison of GEO with state-of-the-art solvers. While in Secs. IIA and IIB we implemented and fine-tuned each solver, in Sec. IIC, we leverage the state-of-the-art results from nine other solvers reported in the literature in the last two decades. In the latter case, each non-GEO solver was thoroughly fine-tuned by the researchers of each reference. This portfolio optimization problem is so canonical that when a new solver is proposed, researchers can compare their results by taking the results from the new proposed solver, as long as the benchmark problems are run in identical conditions. This was one of the main motivations for us to choose this well-established benchmark problem. The "rules of the game" for reporting each market index and performance indicator are reported in Appendix A 2. In contrast, for the other two subsections, the criteria of evaluation are different, and it emphasizes the performance of GEO when one imposes a limit on the total wall-clock time (Sec. IIA) and when there is a limited number of calls to the cost function (Sec. IIB). The latter is a potential scenario when the bottleneck or expensive step is the cost function evaluation itself (e.g., as it is the case of drug discovery where each evaluation (each candidate molecule) might require synthesis in the lab and an expensive and long process towards its Food and Drug Administration (FDA) approval).

To the specific question of (G,T,S) the Reviewer is correct that the authors from the original reference back in 2000 ran each strategy (Genetic algorithms, Tabu search and Simulated annealing) and reported their best results for each. Since this paper was one of the first ones adopting the current metrics and benchmarking procedure, when the community started creating more sophisticated metaheuristic strategies, they continue referring to the results in this paper as GTS, implying the values correspond to the best result from either G, T, or S

Revisions to the manuscript: To clarify this to the reader, we have emphasized further in Sec. IIC that the results from the state-of-the-art competitors have been independently fine-tuned and are available in the literature. We have also added more context at the beginning of the Results section to contextualize all the different results subsections and strategies presented in the paper.

-Table I is not so easy to read with the many entries. Is it worth including first 4 optimizers here? They don't seem competitive (no to mention their columns are mostly "-"s)

Reply: We have followed the Reviewer's suggestion and we have simplified the table accordingly (please see new Table I). The main reason for including all the solvers was to follow the format from previous papers which compared to these early metaheuristics from the early 2000's. Since this cardinality constrained problem has been used for benchmarking solved since the late 1990s, these were the solvers which have been competitive since the last two decades or so ago. As time has passed, and in particular in the last decade, other researchers have proposed other figure of merits to assess the performance of the solvers. Since these were not known in the first papers, they have been conventionally included in the most recent papers with the '-' since the data is not available. Others have adopted these other metrics as valuable and that is the reason the more recent papers all evaluate them, and we decided to include them here as well.

The original intention from having the complete table was to have the whole spectrum of solvers and help position our quantum-inspired GEO in the historical progress of optimizers for this specific task, clearly showing GEO outperforms these solvers from the early 2000's up to a decade ago, and being on par with the current state-of-the-art optimizers. Note that with the suggestion from the other Reviewer to include a classical version of GEO, called NADE-GEO in the revised version, we can see that NADE-GEO is competitive with solvers from about a decade ago, but not as competitive as TN-GEO or the new solvers.

Revisions to the manuscript. We have collapsed the table appearing in the main text as suggested by the reviewer. The original table, expanded now with the NADE-GEO results, has been moved to Appendix C.

-RCABC appeared to perform comparably to TN-GEO, it would be nice to see more discussion of this or even a more detailed comparison

Reply: We have expanded the discussion on RCABC at the end of the Results section (Sec. IIC). Note in the current version we have replaced the name RCABC to ABC-HP to match the name of the best performing variant in that 2021 paper cited, and which corresponds to the results reported here.

References:

- [1] Bello, I., Pham, H., Le, Q.V., Norouzi, M. and Bengio, S., Neural combinatorial optimization with reinforcement learning. *arXiv preprint arXiv:1611.09940* (2016).
- [2] Bengio, Y., Lodi, A. and Prouvost, A., Machine Learning for Combinatorial Optimization: a Methodological Tour d'Horizon. CoRR abs/1811.06128 (2018). *arXiv preprint arXiv:1811.06128* (2018).
- [3] Banner, W.P., Hadiashar, S.B., Mazur, G., Menke, T., Ziolkowski, M., Kennedy, K., Romero, J., Cao, Y., Grover, J.A. and Oliver, W.D., Quantum Inspired Optimization for Industrial Scale Problems. *arXiv preprint arXiv:2305.02179* (2023).
- [4] Lopez-Piqueres, J., Chen, J. and Perdomo-Ortiz, A., Symmetric tensor networks for generative modeling and constrained combinatorial optimization. *Machine Learning: Science and Technology*. 4, 035009 (2023).

- [5] Rudolph, M.S., Miller, J., Motlagh, D., Chen, J., Acharya, A. and Perdomo-Ortiz, A., Synergy between quantum circuits and tensor networks: Short-cutting the race to practical quantum advantage. *arXiv preprint arXiv:2208.13673* (2022).
- [6] Hibat-Allah, M., Mauri, M., Carrasquilla, J. and Perdomo-Ortiz, A., A Framework for Demonstrating Practical Quantum Advantage: Racing Quantum against Classical Generative Models. *arXiv preprint arXiv:2303.15626* (2023).
- [7] Uria, B., Côté, M.A., Gregor, K., Murray, I. and Larochelle, H., Neural autoregressive distribution estimation. *The Journal of Machine Learning Research*, 17(1), pp.7184-7220 (2016).
- [8] Ponce, M., Herrman, R., Lotshaw, P.C., Powers, S., Siopsis, G., Humble, T. and Ostrowski, J., Graph decomposition techniques for solving combinatorial optimization problems with variational quantum algorithms. *arXiv preprint arXiv:2306.00494* (2023).
- [9] Ushijima-Mwesigwa, H., Shaydulin, R., Negre, C.F., Mniszewski, S.M., Alexeev, Y. and Safro, I., Multilevel combinatorial optimization across quantum architectures. *ACM Transactions on Quantum Computing*, 2(1), pp.1-29 (2021).
- [10] Zhou, Z., Du, Y., Tian, X. and Tao, D., QAOA-in-QAOA: solving large-scale MaxCut problems on small quantum machines. *Physical Review Applied*, 19(2), p.024027 (2023).

Reply to Reviewer #2

**** Key results ****

This paper proposes to use a quantum inspired generative model to help more efficiently explore the space of feasible solutions for combinatorial optimization problems. The generative model is used to sample new candidate solutions: from a set of already explored solutions with their associated cost, the generative model is trained to learn a distribution over the solution space for which the probability of each seen solution is proportional to their associated cost, thus potentially making it possible to sample new promising candidate solutions. The authors focus on using a quantum inspired generative model based on tensor networks (matrix product states), which have been previously introduced in [15].

*The proposed approach is evaluated on the task of portfolio optimization and compared with state of the art optimization algorithms for this task. **The experiments reveal that the approach is competitive with these state of the art solvers** (which have been fine tuned for decades), even outperforming some of them on this particular task.*

**** Validity ****

The results presented in the paper are valid. Both the methodology and the experiment setup is sound.

**** Data & methodology ****

*The paper is overall clearly written **and the experiments demonstrate well the potential benefits and usefulness of the method.***

Reply: We thank the referee for the very positive feedback on our proposed framework. Below we address the concerns raised in the comments.

One concern that I have is that the advantage of using a quantum approach is not clearly demonstrated. In particular, it is not clear how the fact that quantum inspired models were chosen for the generative part is key to obtaining the experimental results. At the very least I believe the proposed approach should be compared with replacing the MPS model with a simple HMM learned using the Baum-Welch (i.e. EM) algorithm. Other generative models should be considered as baselines as well (e.g. a simple RNN trained using backpropagation through time or a more complex model such as NADE [Uria et al]). It may be the case that replacing the MPS model by such an alternative non-quantum inspired model would lead to similar result. To sum up, I believe the authors should experimentally investigate and discuss more in depth to which extent the quantum part of their approach is necessary and beneficial.

*Uria, Benigno, et al. "Neural autoregressive distribution estimation." *The Journal of Machine Learning Research* 17.1 (2016): 7184-7220.*

Reply: We agree with the Reviewer this is a critical point that deserves further study since it is in general an open question. One of the highlights for GEO is the flexibility of swapping classical, quantum inspired, and quantum models in the future as quantum technologies mature. We have followed the suggestion from the Reviewer and implemented NADE, which is the most complex model of the ones suggested. In the new results summarized in Table V (computed from the explicit values in Table III), it can be seen that although NADE-GEO is a competitive solver, it is only statistically on par with the solvers which were the best performers up to about a decade ago (GTS, IPSO, IPSO-SA, and PBILD), but statistically different from its quantum-inspired version (TN-GEO), and the solvers proposed from 2015

and onwards (GRASP, ABCFEIT, AAG, VNSQ, and ABC-HP). Although this does not constitute a proof in any way of the superiority of quantum or quantum-inspired models, this corresponds to the experimental investigation which hints to the value and promising directions we highlight in this work from this flexible framework.

To the milestone of understanding when quantum or quantum-inspired models could be key to obtaining an advantage over classical models, several subsequent publications have attempted to address this point. For example, in Ref. [1], we have recently published a version of TN-GEO which uses symmetries which are relatively easy in these quantum-inspired TN models. We showed how the search space of problems with equality constraints can be represented efficiently in quantum-inspired generative models, but this is not necessarily the case for classical deep-neural-network-based generative models due to their non-linear activation units. That recent work poses a more concrete differentiator between the capabilities of quantum-inspired models with respect to traditional deep learning generative models. In general, the question of practical quantum advantage in quantum machine learning is still wide open, but in this recent work, Ref. [2] below, the proposed framework there leverages our GEO proposal to establish a clear-cut criteria from optimization problems to concretely test the performance of both classical and quantum generative modeling proposals. From a practical point of view, and in case of real and relevant real-world applications, this question would need to be approached on a case-by-case basis, but that subsequent paper proposes a framework where the performance of quantum generative models can be tested, following similar metrics inspired from this work.

**** Appropriate use of statistics and treatment of uncertainties ****

Yes, the results are reported appropriately using classical statistical tools and treatment of uncertainties.

Reply: We thank the referee for the positive feedback on our statistical analysis.

**** Conclusions: Do you find that the conclusions and data interpretation are robust, valid and reliable? ****

Overall the conclusions and interpretation of the experimental results are robust and reliable. Though, as I mentioned above, I believe there is a set of experiments missing which would demonstrate more the relevance of a quantum approach to the generative model part of the proposed method.

Reply: We thank the referee for the positive feedback on our overall conclusions. We hope the new numerical experiments, including the simulations of NADE-GEO strengthens our claims around the potential benefit of quantum inspired generative models, and in general, of quantum models as an alternative to be explored in the near term.

**** References ****

The manuscript references previous literature appropriately.

Reply: We thank the referee for the positive feedback on our literature coverage.

**** Clarity and context ****

The paper is very well written and structured, and easy to follow for someone familiar with tensor network models.

Reply: We thank the referee for the positive feedback on the content and the structure of the paper.

**** Suggested improvements ****

- Include an experiment to compare the proposed approach with non-quantum generative models

Reply: Following the advice from the referee, we have implemented NADE-GEO and the comparison with TN-GEO and the other SOTA solvers are included throughout Tables I-V. As previously mentioned, the classical ML model is competitive in the sense that is comparable with models from about a decade ago. But still underperforming TN-GEO, which is on par with the state-of-the-art solvers for this application. This is also included in the new summary Table II in the main text, where it can be seen the superior performance of TN-GEO over NADE-GEO in this specific application.

- It could be a nice addition to have experiments on another combinatorial optimization problem than portfolio optimization.

- Related to the previous point, the authors may consider presenting their approach in a more general context rather than specifically for the portfolio optimization problem. As I understood it, the proposed approach can be applied to many different kind of optimization problems but the current presentation can suggest that the method is tailored specifically for portfolio optimization. A more general presentation of the method, as well as a clear explanation of the different kind of problems the approach can be applied to, could make for a more impactful paper (by reaching a wider audience). For example, can the approach be applied to any MIP?

Reply: We agree with the reviewer that the current presentation hints to GEO as a solution tailored only for the portfolio optimization problem, where in reality, as the referee mentioned, GEO is more in the category of a generic solver. Although we had mentioned in a couple of places its black-box nature, many readers might not immediately associate this concept with a solver that can deal with any cost function, and therefore any combinatorial problem. This black-box feature is one of the most prominent ones which render our approach advantageous compared to other quantum heuristics, such as the quantum approximate optimization algorithm (QAOA), which relies on the cost function to be a polynomial in terms of the binary variables. To mitigate missing these essential points raised by the referee, the introduction has been significantly revised and restructured adding clarity to the main features of our framework.

Although we did not explore explicitly in this work the case of problems containing both discrete and continuous variables (e.g., MIP) or only continuous variables, we discussed in the Outlook how an approach as the one in Ref. [3] could allow to generalize to these mixed variable cases by using such hybrid quantum-classical generative models within GEO. In a recent publication, Ref. [4], our team addressed this question of how to treat generative models with continuous variables (or a mixture of discrete and continuous) directly with TN models, which would be directly applicable to MIP and other related problems.

To the question of including other applications, the main motivation for selecting this specific instantiation of portfolio optimization was the availability of concrete benchmarks and an extensive literature of solvers which have been fine tuned for over the past decades. In particular, we wanted to make sure we could make stronger claims in a well-established benchmark problem, and assess in this

way where this new quantum-inspired solution TN-GEO fits within the suite of SOTA algorithms. Although this could be done for other well-known benchmarking problems, we felt the paper was extensive enough since it presents the introduction of this new framework plus demonstrating some of its highlights reflected in the three Results subsections: 1) the possibility to leverage data observed with other solvers (Sec. IIA), 2) its high performance in the regime of very limited calls to the cost function evaluation (Sec. IIB) and 3) its performance compared to SOTA algorithms under the rules established by those specific benchmarks (Sec. IIC). Although we considered extending to other application domains is outside of the scope of this work, other recent independent works have considered other real-world applications of GEO. For example, in Ref. [5], the authors considered an industrial case related to a floor planning NP-hard problem. In Ref. [6] the authors considered quantum-inspired generative models similar to the ones we propose here, to explore the solution space of candidates in molecular discovery. We hope the referee considers as well that the work is complete as is, given the new developments supporting the range of applications of our work.

Revisions to the manuscript: We hope the new Introduction brings more clarity to the essential points raised by the referee. To emphasize the point from the referee that even MIP or problems with continuous variables could be addressed within GEO, we have added some comments in the Outlook about recent work extending and addressing this challenge, and how it fits within our framework.

References:

- [1] Lopez-Piqueres, J., Chen, J. and Perdomo-Ortiz, A., Symmetric tensor networks for generative modeling and constrained combinatorial optimization. *Machine Learning: Science and Technology*. 4 035009 (2023)
- [2] Hibat-Allah, M., Mauri, M., Carrasquilla, J. and Perdomo-Ortiz, A., A Framework for Demonstrating Practical Quantum Advantage: Racing Quantum against Classical Generative Models. *arXiv preprint arXiv:2303.15626* (2023).
- [3] Rudolph, M.S., Toussaint, N.B., Katarawa, A., Johri, S., Peropadre, B. and Perdomo-Ortiz, A., Generation of high-resolution handwritten digits with an ion-trap quantum computer. *Physical Review X*, 12(3), p.031010 (2022).
- [4] Meiburg, A., Chen, J., Miller, J., Tihon, R., Rabusseau, R., Perdomo-Ortiz, A. Generative Learning of Continuous Data by Tensor Networks. *arXiv preprint arXiv: 2310.20498* (2023).
- [5] Banner, W.P., Hadiashar, S.B., Mazur, G., Menke, T., Ziolkowski, M., Kennedy, K., Romero, J., Cao, Y., Grover, J.A. and Oliver, W.D., Quantum-Inspired Optimization for Industrial Scale Problems. *arXiv preprint arXiv:2305.02179* (2023).
- [6] Moussa, C., Wang, H., Araya-Polo, M., Bäck, T. and Dunjko, V., 2023. Application of quantum-inspired generative models to small molecular datasets. *arXiv preprint arXiv:2304.10867* (2023).

REVIEWERS' COMMENTS

Reviewer #1 (Remarks to the Author):

The authors have considered and reasonably addressed the comments of the referees, resulting a significant number of changes and improvements to the manuscript. Hence I am happy to recommend for publication.

Reviewer #2 (Remarks to the Author):

I am overall satisfied by the answers from the authors to my initial review and to the consequent revisions that have been made to the manuscript.

I find that stylistically speaking, the quality of the introduction has decreased in some of the new paragraphs added in the revision (see detailed comments below). I also give below some relatively minor stylistic comments about the new content that has been added to the revision:

p. 1, second item: "***either*** from previous..." there is a missing "or" and second part of the sentence.

p. 2 first item: This open -> This opens

p. 2 The last item is stylistically very odd and reads more than a rebuttal to the first round of reviews than an actual paragraph from an introduction. More precisely, this list is presented as a list of "salient highlights" of the proposed approach; is "Nothing is special about the portfolio optimization problem" a "salient highlight" of the contribution. I understand the aim but this is done in a very clumsy way, in my opinion. I would suggest rewriting this item and rephrasing in the direction of emphasizing the "versatility" of the method, while justifying the focus on portfolio optimization.

p.2 last item: "... *as* the NP hard problem *as* the workhorse..." is a bit stylistically clumsy

p.2 last item: line starting with a comma after "Ref [8]"

p.6 to be consistent, the item 10)'s sentence in the before last paragraph should end with
";" instead of ".'

p. 10 before last paragraph of first column: "it is to use it as" -> "is to use it as"

p.10 "DATA AVAILABILITY": "have been deposited" -> e.g. "is available"

p.14 sec 4: "introduced by Uria [37]" -> "introduced by Uria et al. [37]"

"is that it models" -> "is to model"

Reply to Reviewer #1

We thank Reviewer #1 for their feedback throughout this review process and for their recommendation to publish this significantly revised version.

Reply to Reviewer #2

We thank Reviewer #2 for their recommendation to accept this latest version for publication. In this resubmission, we have incorporated all the stylistic changes suggested by the reviewer. We thank the reviewer for this valuable feedback throughout this review process.